# Relationship between Post-Stroke Cognitive Impairment and Severe Dysphagia: A Retrospective Cohort Study

**DOI:** 10.3390/brainsci12060803

**Published:** 2022-06-19

**Authors:** Jia Qiao, Zhi-Min Wu, Qiu-Ping Ye, Yong Dai, Zu-Lin Dou

**Affiliations:** 1Department of Rehabilitation Medicine, the Third Affiliated Hospital of Sun Yat-sen University, No. 600, Tianhe Road, Tianhe District, Guangzhou 510630, China; qiaoj5@mail2.sysu.edu.cn (J.Q.); qiuxy1008@163.com (Q.-P.Y.); 2Department of Neurosurgery, the Third Affiliated Hospital of Sun Yat-sen University, Guangzhou 510630, China; wuzhm28@mail2.sysu.edu.cn; 3Clinical Medical College of Acupuncture-Moxibustion and Rehabilitation, Guangzhou University of Chinese Medicine, No. 232, Waihuaneast Road, Panyu District, Guangzhou 510006, China; doctordaiyong2018@163.com

**Keywords:** post-stroke cognitive impairment (PSCI), severe post-stroke dysphagia (PSD), Videofluoroscopy Swallowing Study (VFSS), Penetration-aspiration Scale (PAS), Functional Oral Intake Scale (FOIS), retrospective study

## Abstract

*Objective*: To investigate the relationship between post-stroke cognitive impairment (PSCI) and severe post-stroke dysphagia (PSD) and explore the risk factors related to PSCI combined with severe PSD. *Methods*: Data from patients were collated from the rehabilitation-specific disease database. The Mini-Mental State Examination (MMSE), Montreal Cognitive Assessment (MoCA), Videofluoroscopy Swallowing Study (VFSS), Penetration-aspiration Scale (PAS), and Functional Oral Intake Scale (FOIS) were used to evaluate cognitive and swallowing functions. Differences between groups were determined by the Pearson chi-square test (χ^2^) or Fisher exact test. PAS and FOIS data were analyzed with the use of the Wilcoxon rank-sum or Kruskal–Wallis test in the prespecified subgroup analysis. Risk factors were investigated by multivariate logistic regression. *Results*: A total of 1555 patients were identified with PSCI. The results indicated that patients with PSCI had a higher incidence rate of severe PSD as compared to patients without PSCI (*p* < 0.001). Patients with severe PSCI were more likely to clinically manifest oral phase dysfunction (*p* = 0.024), while mild PSCI patients mainly manifested pharyngeal phase dysfunction (*p* < 0.001). There was a significant difference in FOIS score changes between subgroups during the hospitalization period (severe PSCI vs. moderate PSCI and severe PSCI vs. mild PSCI) (all *p <* 0.001). In addition, multivariate logistic regression revealed pneumonia (*p* < 0.001), tracheotomy (*p* < 0.001), and dysarthria (*p* = 0.006) were related to PSCI, combined with severe PSD. *Conclusion*: PSCI may be related to severe PSD. Patients with severe PSCI were more likely to manifest oral phase dysfunction, while mild PSCI manifested pharyngeal phase dysfunction. Pneumonia, tracheotomy, and dysarthria were risk factors related to PSCI combined with severe PSD.

## 1. Introduction

Post-stroke cognitive impairment (PSCI) is one of the common but serious complications after stroke, with an overall estimated prevalence of about 30% [1,2,3]. The cognitive impairment is generally classified as subjective cognitive decline, mild cognitive impairment, and dementia [4]. Since cognitive function mainly includes memory, executive function, attention, language, and visuospatial ability, cognitive impairment usually refers to the impairment of one or more of these listed functions [5]. The common assessment tools for PSCI include Mini Mental State Examination (MMSE), Montreal Cognitive Assessment (MoCA), and Cognitive Assessment for Stroke Patients (CASP). For patients with language impairment, the efficiency of CASP is higher than MMSE and MoCA [6]. It is noteworthy that language impairment can be an isolated symptom after stroke, which is associated with damage of the language centers [7]. For example, in the case of a focal brain lesion in the dominant hemisphere, the patient may present with aphasia (language disorder), which is not classified as cognitive impairment.

Post-stroke dysphagia (PSD), another common complication after stroke, is closely associated with aspiration, pneumonia, malnutrition, and dehydration, with a high estimated prevalence in the first week [8,9,10,11,12,13,14,15]. Patients with severe PSD often need gastric tubes to assist in swallowing for a long time. Swallowing is a complex process that requires the coordination of different brain regions, including the central pattern generator of the medulla oblongata, the frontal cortex, subcortex, etc. [16,17]. However, cognitive functions are controlled by these cortex regions as well [18]. Given that the cortex regions associated with the cognitive and swallowing functions are overlapped, some researchers believe that PSCI is associated with the severity of PSD and should be treated as a predictive indicator [16,19,20,21]. However, this viewpoint does not appear to be widely recognized [22,23,24,25]. During a long period after stroke, many patients suffered from both PSCI and PSD in clinical settings. As compared to those only with PSD, patients with both complications were more likely to obtain a worse prognosis [26,27].

Currently, there is insufficient evidence in the medical literature to draw definite conclusions about the correlation between PSCI and severe PSD. Therefore, we hypothesized that there was a close association between them. To examine this, we first explored and assessed the impacts of PSCI on the occurrence, performance, and treatment outcomes of severe PSD. Second, we further investigated risk factors of PSCI complicated with severe PSD, which may potentially provide help for clinical evaluation and treatment.

## 2. Method

### 2.1. Study Design and Subjects

This is a retrospective study with data extracted from the cohort database of the Third Affiliated Hospital of Sun Yat-Sen University. Patients were included in this database according to the diagnosis at discharge from the rehabilitation department, and the details provided included medical records, auxiliary examinations, and treatment processes. In this database, Mini-Mental State Examination (MMSE) or Montreal Cognitive Assessment (MoCA) was the measurement tool of cognitive function, while Videofluoroscopy Swallowing Study (VFSS), Penetration-aspiration Scale (PAS), and Functional Oral Intake Scale (FOIS) were the main measurements of swallowing functions. In the present study, we primarily reviewed patients who were newly diagnosed with stroke from 2010 to 2021. The inclusion met ethical standards of ethical committees of the Third Affiliated Hospital of Sun Yat-sen University (02-351-01), and an exemption to informed consent requirements was granted. 

### 2.2. Inclusion and Exclusion Criteria

Inclusion criteria were: (1) age > 18 years; (2) diagnosed with stroke (including all hemorrhagic and ischemic stroke, e.g., subdural hematoma, subarachnoidal hemorrhage, intracerebral hemorrhage and ischemia, brainstem strokes, etc.) by CT/MRI and clinical symptoms between two weeks and six months; (3) cognitive function was evaluated at admission by Mini-Mental State Examination (MMSE) or Montreal Cognitive Assessment (MoCA); (4) swallowing function was evaluated by special methods (VFSS, PAS, and FOIS) during hospitalization, only FOIS < 4 was included. Exclusion criteria were: (1) incomplete case information; (2) the assessment of cognitive and swallowing function was not given during hospitalization; (3) patients with other cerebral diseases (e.g., encephalitis, psychosis, intracranial space-occupying lesions, psychosis, etc.); (4) patients with fluent aphasia unable to complete the MMSE or MoCA assessment; (5) patients with cognitive impairment already impaired prior to the onset of stroke.

### 2.3. The Assessment Methods 

#### 2.3.1. Cognitive Impairment

The diagnosis of cognitive impairment was mainly made according to the history of present illness and the use of one or more validated instruments, including MMSE (Score range: 0–30; 21–24, mild cognitive impairment; 13–20, moderate cognitive impairment; 20–21, severe cognitive impairment; 3-point change considered clinically significant), MoCA (Score range: 0–30, higher score equaling less cognitive impairment) [4]. An MMSE < 27 or MoCA < 26 was considered cognitive impairment in the present study [28,29].The assessment above was given after admission to the rehabilitation department by a well-trained clinician. Subgroups of the present study were divided according to the severity of PSCI.

#### 2.3.2. Swallowing Function

FOIS, VFSS, and PAS were adopted to assess the swallowing functions by the well-trained speech-language pathologist (SLP). FOIS was used to assess dietary intake, and a score of less than four was considered severe dysphagia in the present study [18,30]. FOIS was evaluated at admission and at discharge to assess the curative effect. VFSS is a dynamic X-ray procedure performed using a fluoroscope and recorded in a video format. The VFSS examination was evaluated at admission by the same SLP. During the VFSS, radiation exposure ranged from the top of the nasal cavity, down to the C7 cervical spine, to the anterior and posterior of the lips, and back of the neck. Patients received 3 mL, 5 mL, and 10 mL of thickened and diluted barium liquid (contrast media: 60% *w/v* barium sulfate suspension), and the steps were performed according to the modified Logemann protocol [31]. The video was analyzed by the above SLP. The various phases of swallowing and PAS score were recorded [32]. The phases of swallowing, including the oral phase (oral preparatory phase and oral propulsive phase), pharyngeal phase, esophageal phase, dysfunction existing at any of the four phases of the physiologic swallow [31,33,34,35,36], and the definition of different phases are shown in Table 1 and Figure 1. PAS was also scored by VFSS, and the highest score was recorded during the examination. PAS is an 8-point scale used to determine dysphagia severity. The higher the score, the more severe the penetration and aspiration symptoms [37]. 

#### 2.3.3. Risk Factors

Gender, age, smoking history, drinking history, underlying diseases (like pneumonia, dysarthria), history of gastric tube and tracheotomy, etc., were fit as the independent variables to identify the potential relationship between PSCI and severe PSD. 

### 2.4. Data Analysis

Differences between groups were determined by the Pearson chi-square test (χ^2^) or Fisher exact test. PAS and FOIS data were analyzed with the use of the Wilcoxon rank-sum or Kruskal–Wallis test in the prespecified subgroups analysis. Risk factors were first analyzed with descriptive statistics, then those with a *p*-value less than 0.05 were further investigated by multivariate logistic regression. All Statistical analyses were performed using SPSS software (version 23.0, SPSS/IBM, Armonk, NY, USA). Statistical charts were accomplished using GraphPad Prism (version 9.0, Graphpad Software Inc., San Diego, CA, USA).

## 3. Results

### 3.1. Relationship between PSCI and Severe PSD

A total of 9144 stroke patients were screened from the database, of which 1555 had PSCI, including 839 (54%) PSCI combined with severe PSD, 716 (46%) PSCI with non-severe PSD; and 7589 had no PSCI, including 2314 (30.5%) with severe PSD, and 5275 (69.5%) without PSCI or PSD. The Pearson chi-square test showed that patients with PSCI had a higher incidence rate of severe PSD compared to patients without PSCI, which was statistically different (χ^2^ = 314.5, *p* < 0.001) (Figure 2A). 

Patients with incomplete medical records were excluded. Finally, a total of 331 were included for the analysis of risk factors, and 154 patients were included for further subgroup analysis based on the severity of PSCI. 

The clinical characteristics of PSCI are shown in Table 2. There was a significant difference in age (*p* = 0.031), smoking history (*p* = 0.023), drinking history (*p* = 0.002), pneumonia history (*p* < 0.001), stroke site (*p* < 0.001), gastric tube history (*p* < 0.001), tracheotomy history (*p* < 0.001), and dysarthria history (*p* = 0.007) between PSCI combined with severe PSD and PSCI without severe PSD group (Table 2). 

### 3.2. Subgroup Analysis Stratified by the Severity of PSCI

Among 154 patients with PSCI and severe PSD, 63 were involved in the mild PSCI subgroup, 28 in the moderate PSCI subgroup, and 63 in the severe PSCI subgroup. The results showed that the FOIS scores were significantly higher at discharge than at admission in three subgroups (*p* < 0.001) (Figure 2B). Additionally, at admission, the FOIS score in the severe PSCI subgroup was higher than in the mild PSCI subgroup (*p* < 0.01), and the moderate PSCI subgroup score was also higher than in the mild PSCI subgroup (*p* < 0.05) (Figure 2B). Furthermore, the FOIS score change in the severe PSCI subgroup was lower than moderate PSCI subgroup (*p* < 0.001) and mild PSCI subgroup (*p* < 0.001) during hospitalization (Figure 2C), suggesting that the mild and moderate PSCI subgroups obtained better curative effects than the severe subgroup during hospitalization. There was no statistical difference among the three groups in PAS score (*p* > 0.05) (Figure 2D). 

Among the mild PSCI, moderate PSCI, and severe PSCI subgroups, the results demonstrated that there were statistical differences in oral phase (*p* = 0.024), pharyngeal phase (*p* < 0.001), oral+pharyngeal phase (*p* = 0.022), and oral + pharyngeal + esophageal phase (*p* = 0.048) (Table 3).

### 3.3. Risk Factor Analysis of PSCI with Severe PSD


The descriptive statistics revealed that there were significant differences in age, smoking, drinking, pneumonia, stroke site, lesion side, gastric tube, tracheotomy, and dysarthria (*p <* 0.05, Table 2). These factors were further included in the regression model, and the logistic regression analysis suggested that the history of pneumonia (*p <* 0.001, OR = 16.9, 95%CI 5.3–53.3), tracheotomy (*p <* 0.001, OR = 34.5, 95%CI 9.5–126.0), and dysarthria (*p <* 0.01, OR = 4.6, 95%CI 1.5–13.4) were related to patients with PSCI and severe PSD (Figure 2E).

## 4. Discussion

The main finding of this retrospective study was that there might exist correlation between PSCI and severe PSD. Patients with severe PSCI were more likely to manifest oral phase dysfunction, while mild PSCI manifested as pharyngeal phase dysfunction. The clinical efficacy of swallowing therapy for patients with severe PSCI may be worse than for those without. Moreover, pneumonia, tracheotomy, and dysarthria were risk factors for PSCI complicated with severe PSD. 

### 4.1. The Relationship between Cognitive and Swallowing Functions

Regarding the relationship between cognitive and swallowing functions, we performed analyses and found that the PSCI was related to the occurrence of severe PSD. Similar findings could be found in the results of other analyses [16,19,20] which supported the validity of our original work. Swallowing behavior is not only controlled by the brainstem, but also by higher-level nerve centers including the frontal cortex, insula, anterior cingulate gyrus, etc., especially the dorsolateral prefrontal cortex [38,39]. Coincidentally, cognitive functions also involve the cerebral cortex including the frontal lobe [18], suggesting there may exist an intersection in the neural network between swallowing behavior and cognitive function [40]. However, the mechanisms by which swallowing changes relate to cognitive impairment remain quite complex and unclear. Some studies found that the brain regions related to swallowing are easily damaged in the pathological process of Alzheimer’s disease, which may be the main cause of dysphagia frequently combined with cognitive impairment [17,39,41]. Besides, multiple strokes involving periventricular white matter can also lead to both vascular dementia and dysphagia at the same time, which also supports this phenomenon [42]. Therefore, for patients with PSCI, screening, and assessment of swallowing function are necessary. 

In the subgroup analysis, we found that the oral phase of swallowing was related to severe PSCI, while patients with mild PSCI were more likely to manifest as pharyngeal phase dysfunction, which was consistent with one former study [20]. During the oral preparatory phase, the masticatory muscles for jaw closing and opening are involved in stabilizing the mandible during oral closure and tongue elevation [35]. The severe PSCI may mainly interfere with the preparatory phase of swallowing, including ingestion and harsh processing of food in the oral cavity [40,41]. Moreover, the patients with severe PSCI had poor treatment efficacy during hospitalization, most likely because those without executive dysfunction could cooperate well with therapists. Thus, severe PSCI, especially executive function, may be an important factor that affected swallowing therapy for patients. 

It is noteworthy that some earlier studies considered there to was no association between cognitive functioning and dysphagia after stroke [22,23,24,25], one possible reason for this being differences in methodology. For example, one of these studies used the Outcome Severity Scale (DOSS) as the evaluation of the swallowing function [24], while the FOIS, PAS, and VFSS were applied as the measurement in the present study. Furthermore, telephone interviews were used to assess the cognitive function of patients by Meade et al. [22], while the MMSE/MoCA was finished face to face in the present study. This difference should not be neglected when interpreting the results.

### 4.2. Risk Factor Analysis of PSCI with Severe PSD 

The study found that patients with a history of pneumonia, tracheotomy, and dysarthria presented the risk factors for PSCI with severe PSD. First, patients with lung diseases may have negative impacts on the breath-swallowing coordination pattern, leading to the occurrence of PSD [43]. Second, a history of tracheotomy may increase the incidence of dysphagia due to the invasive operation, possibly causing damage to the muscles and nerves that control the swallowing process [44] and causing changes in the biofluid mechanics in the trachea, consequently resulting in changes in the pharyngeal phase [45]. Third, dysarthria has proven to be an important predictor of dysphagia and helpful in the therapeutic process of swallowing problems [46,47]. This is related to paralysis, weakness, or incoordination of the muscles in the mouth, face, and tongue, which are particularly essential for the swallowing process. Muscles of articulation and swallowing are controlled by the motor cortex related to voluntary muscular movements in the frontal lobe [47]. Thus, there is nothing surprising regarding dysarthria as a risk factor for PSCI with severe PSD in this study.

### 4.3. Limitations

There are some limitations to this retrospective study. First, only 331 patients with complete medical records were included in the analysis for risk factors, which may lead to bias in the results. Second, the swallowing function of patients with PSD probably recovers spontaneously in the early stage of stroke, which may influence the results [11]. Third, the present study only included severe PSD patients for further analysis, while mild and moderate PSD patients were not included due to the limited data. In summary, the analysis and interpretation of the results should be considered cautiously, and more high-quality prospective studies are urgently needed. 

## 5. Conclusions

Severe PSD may be related to PSCI. Patients with severe PSCI are more likely to manifest oral phase dysfunction, while mild PSCI manifests as pharyngeal phase dysfunction. Clinical efficacy for patients with severe PSCI may be worse than those without. Pneumonia, tracheotomy, and dysarthria were risk factors related to severe PSD combined with PSCI. The mechanism of severe PSCI on swallowing dysfunction remains unclear, and further studies are needed to verify it.

## Figures and Tables

**Figure 1 brainsci-12-00803-f001:**
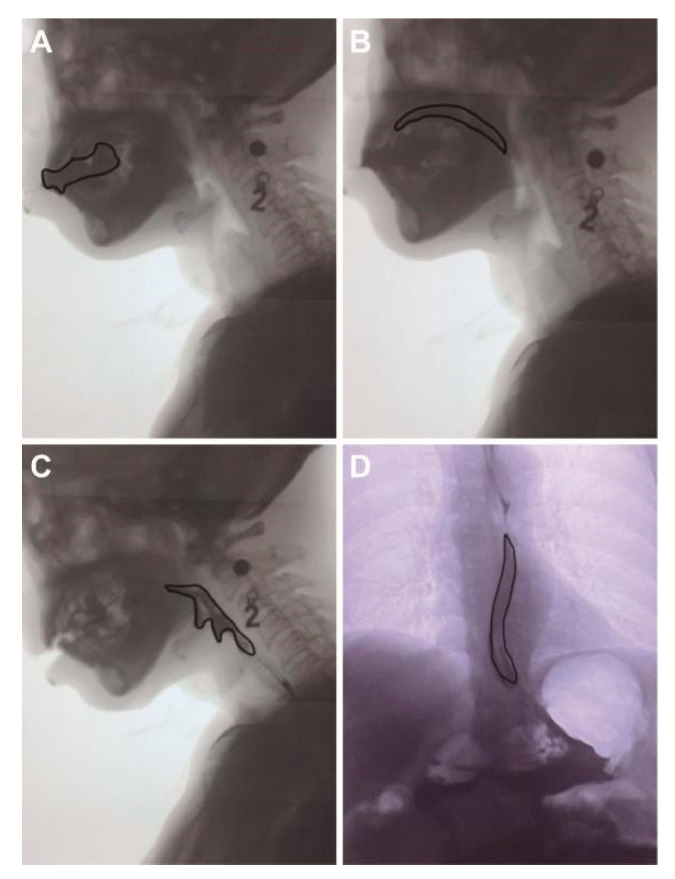
The four phases of swallowing for a 55-year-old male patient under standard Videofluoroscopy Swallowing Study. Radio-opaque contrast was drunk by the patient and dynamic changes of contrast were then recorded. (**A**), Oral preparatory phase; (**B**), Oral propulsive phase; (**C**), Pharyngeal phase; (**D**), Esophageal phase. Note: number “2”, oral contrast agent 2 (thickened liquid).

**Figure 2 brainsci-12-00803-f002:**
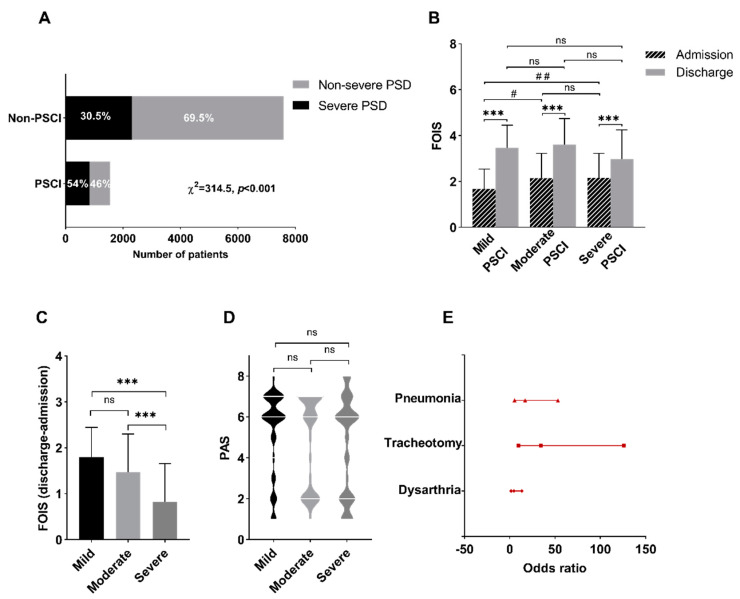
(**A**) The proportions of severe PSD and PSCI. (**B**) The Functional Oral Intake Scale (FOIS) scores of patients in mild, moderate, and severe PSCI subgroups. FOIS scores at discharge were significantly higher than scores at admission for patients among three groups (*p* < 0.001), suggesting almost all patients obtained effective care and rehabilitation. (**C**) The FOIS score change in the severe PSCI group was lower than in the moderate PSCI (*p* < 0.001) and mild PSCI groups (*p* < 0.001) during hospitalization, suggesting that the mild and moderate PSCI subgroups obtained better curative effects than the severe subgroup during hospitalization. (**D**) The Penetration−Aspiration Scale (PAS) scores of patients. The PAS scores among mild, moderate, and severe PSCI subgroups. The results demonstrated that there was no statistical difference between groups in PAS score (*p* > 0.05). (**E**) Multivariate logistic regression analysis for the risk factors related to dysphagia combined with cognitive impairment. The pneumonia, tracheotomy, and dysarthria were demonstrated as risk factors. Note: PSD, post−stroke dysphagia; PSCI, post−stroke cognitive impairment; FOIS, Functional Oral Intake Scale; PAS, Penetration−aspiration Scale; 95%CI, 95% confidence interval; *p* < 0.05 for significance; ns, not significant; *** *p* < 0.001, # *p* < 0.05, ## *p* < 0.01.

**Table 1 brainsci-12-00803-t001:** The start and end point of different phases of swallowing.

Phases of Swallowing	Start Point	End Point
Oral preparatory phase	The food was processed in the oral cavity	
Oral propulsive phase	The tongue voluntary presses the collected bolus/saliva against the palate	Bolus passing the ramus of the mandible
Pharyngeal phase	Bolus passing the ramus of the mandible	The upper esophageal sphincter relaxation
Esophageal phase	The upper esophageal sphincter relaxation	The bolus into the stomach

**Table 2 brainsci-12-00803-t002:** The clinical characteristics of patients with post-stroke cognitive impairment (PSCI).

Variables	PSCI with Severe PSD (n = 154)	PSCI without Severe PSD (n = 177)	*p*
Man, n (%) ‡	114 (74.0)	118 (66.7)	0.183
Age mean (SD) †	65.2 (12.2)	61.9 (15.0)	0.031
History, n (%) ‡			
Smoking	43 (27.9)	31 (17.5)	0.023
Drinking	28 (18.2)	12 (6.8)	0.002
Pneumonia	102 (66.2)	12 (6.8)	<0.001
Comorbidities, n (%) ‡			
Hypertension	111 (72.1)	137 (77.4)	0.265
Hyperlipidemia	24 (15.6)	29 (16.4)	0.843
Diabetes	48 (31.2)	52 (29.4)	0.732
Coronary Heart Disease	15 (9.7)	27 (15.3)	0.133
Stroke type, n (%) ‡			0.400
Hemorrhagic	38 (24.7)	36 (20.3)	
Ischemic	116 (75.3)	141 (79.7)	
Stroke site, n (%) ‡			<0.001
Supratentorial	45 (29.2)	166 (93.8)	
Infratentorial	41 (26.6)	4 (2.3)	
Other	68(44.2)	7(3.9)	
Lesion side, n (%) ‡			<0.001
Left	49 (31.8)	94 (53.1)	
Right	35 (22.7)	61 (34.5)	
Bilateral	50 (32.5)	14 (7.9)	
Other	20 (13.0)	8 (4.5)	
Gastric tube, n (%) ‡	99 (64.3)	12 (6.8)	<0.001
Tracheotomy, n (%) *	27 (17.5)	3 (1.7)	<0.001
Dysarthria, n (%) ‡	110 (62.1)	101 (57.1)	0.007

Note: PSCI, post-stroke cognitive impairment, PSD, post-stroke dysphagia, SD, standard deviation. † Wilcoxon rank-sum test; ‡ Chi-square test; * Fisher exact test; *p* < 0.05 for significance.

**Table 3 brainsci-12-00803-t003:** The swallowing phase outcomes stratified by the severity of PSCI.

No.	Swallowing phase, n (%)	PSCI			*p*
		Mild (n = 63)	Moderate (n = 28)	Severe (n = 63)	
0	Other (none of any items below)	1 (1.6)	1 (3.6)	2 (3.2)	0.802
1	Oral phase	3 (4.7)	3 (10.7)	13 (20.6)	0.024
2	Pharyngeal phase	39 (61.9)	8 (28.6)	16 (25.4)	<0.001
3	Esophageal phase	1 (1.6)	0 (0)	0 (0)	0.483
4	Oral + Pharyngeal phase	16 (25.4)	11 (39.3)	31 (49.2)	0.022
5	Pharyngeal + Esophageal phase	2 (3.2)	2 (7.1)	0 (0)	0.132
6	Oral + Pharyngeal + Esophageal phase	1 (1.6)	3 (10.7)	1 (1.6)	0.048

## Data Availability

The data presented in this study are available on request from the corresponding author. The data are not publicly available due to privacy.

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
