# Peer review of "Relationship between Post-Stroke Cognitive Impairment and Severe Dysphagia: A Retrospective Cohort Study"

_brainsci, 2022, doi:10.3390/brainsci12060803_

Round 1

Reviewer 1 Report

Qiao et al. investigated the association between post-stroke cognitive impairment (PSCI) and post-stroke dysphagia (PDS). They found that patients with PSCI had a significantly higher incidence rate of PSD compared to patients without PSCI. Subgroup analyses revealed that executive dysfunction and a history of dysarthria, tracheostomy, and pneumonia were related to PSD patients with PSCI.

Major comments:

The present study clarified several aspects regarding an association between PSD and PSCI. However, there are many, probably more important aspects, which need further analyses. First, the authors considered functional oral intake scale (FOIS) less than 4 as dysphagia. However, FOIS < 4 relates to varying degrees of non-oral feeding. This means that selected PSD patients suffer from severe dysphagia, and are fed with a feeding tube. Therefore the criteria excluded patients with mild and moderate dysphagia, and caused a selection bias. If the aim of the present study was to test whether the cognitive function is associated with the severity of dysphagia, then patients with mild and moderate dysphagia should be included. Second, there are no assessment regarding cognitive impairment before the onset of stroke. Therefore, it is possible that patients with PSCI have already impaired the cognitive ability before the onset of stoke. Third, patients with mini-mental state examination (MMSE) less than 27 were all included into the PSCI group. It is well known that the prevalence of dysphagia in moderate to severe dementia is high. Therefore, the association between the severity of cognitive impairment and the severity of dysphagia should be evaluated. Forth, the distribution of stroke locations is significantly different between PSCI patients with and without PSD. It is known that the prevalence of dysphagia is different depending on the stroke location (Dysphagia 2017 Dec;32(6):777-784). Therefore, Chi-square test should be conducted to examine whether there was a difference in the prevalence of dysphagia between patients with supratentorial stroke and those with infratentorial stroke.

Minor comments:

  1. Professional English editing throughout the manuscript is required. In particular, lines 107-111 and lines 136-137 do not make sense.

  1. No VFSS and PAS data were shown.

  1. Ref 15 has a typo (fysphagia).

Author Response

Major comments:

The present study clarified several aspects regarding an association between PSD and PSCI. However, there are many, probably more important aspects, which need further analyses.

  1. First, the authors considered functional oral intake scale (FOIS) less than 4 as dysphagia. However, FOIS < 4 relates to varying degrees of non-oral feeding. This means that selected PSD patients suffer from severe dysphagia, and are fed with a feeding tube. Therefore the criteria excluded patients with mild and moderate dysphagia, and caused a selection bias. If the aim of the present study was to test whether the cognitive function is associated with the severity of dysphagia, then patients with mild and moderate dysphagia should be included.

Authors’ response: First, we must apologize for the mistake that we didn't provide an accurate definition of dysphagia. In fact, FOIS < 4 was considered to severe post-stroke dysphagia in the present study. We have clarified this question in the reviewed version of the manuscript.

Second, the real objective of the present study was to investigate the relationship between post-stroke cognitive impairment (PSCI) and severe post-stroke dysphagia (PSD) instead of the relationship between the PSCI and severity of PSD. Therefore, patients with FOIS<4 after stroke were included in present study. In fact, as one of the biggest swallowing rehabilitation centers on China’s mainland, most patients in our center are suffering from severe dysphagia (FOIS<4 and with oral tube feeding), which has resulted in selective bias to some extent. However, we failed to explain this problem clearly in the previous manuscript.

Third, patients with mild and moderate dysphagia should have been included for analysis. As for the question of whether the cognitive function is associated with the severity of dysphagia, we plan to include multicenter data and perform further studies to explore possible results. Jo et al adopted the dysphagia severity scale (DSS) to investigate the relationship between the severity of PSD and PSCI, the results demonstrated that there was a significant relevance between the severity of PSD and PSCI, but the sample size was limited, further prospective study is needed.[1] 

Reference:

[1] Jo SY, Hwang JW, Pyun SB. Relationship Between Cognitive Function and Dysphagia After Stroke. Ann Rehabil Med. 2017 Aug;41(4):564-572.

  1. Second, there are no assessment regarding cognitive impairment before the onset of stroke. Therefore, it is possible that patients with PSCI have already impaired the cognitive ability before the onset of stoke.

Authors’ response: Thank you for your kind reminder. The present study only includes patients with PSCI after stroke, patients with PSCI who have already been impaired before the onset of stroke have been excluded. We have added this detail to the exclusion criteria in the reviewed manuscript (Page8 line 100-101).

  • Third, patients with mini-mental state examination (MMSE) less than 27 were all included into the PSCI group. It is well known that the prevalence of dysphagia in moderate to severe dementia is high. Therefore, the association between the severity of cognitive impairment and the severity of dysphagia should be evaluated.

Authors’ response: Thanks for your suggestion. We further explored the relationship between the severity of PSCI and severe PSD. PSCI patients were divided into mild, moderate, and severe PSCI subgroups. The results have been presented in the result part of the manuscript (Page 8, Line 180-193): Among 154 patients with PSCI+ severe PSD, 63 were divided into mild PSCI subgroup, 28 into moderate PSCI subgroup, and 63 into severe PSCI subgroup. The results showed that the FOIS score was significantly higher before discharge than at admission in three groups (p<0.001) (Fig. 4A). Besides, at admission, the FOIS score in severe PSCI group was higher than mild PSCI group (p<0.01), and the moderate PSCI group was also higher than mild PSCI group (p<0.05). Furthermore, the FOIS score change in severe PSCI group was lower than moderate PSCI group (p<0.001) and mild PSCI group (p<0.001) during hospitalization (Fig. 4B), suggesting that mild and moderate PSCI subgroups obtain better curative effect than severe subgroup during hospitalization. There was no statistical difference among three groups in PAS score (p>0.05).

The results demonstrated that there were statistical differences among mild PSCI, moderate PSCI, and severe PSCI subgroups in oral phase (p=0.024), pharyngeal phase (p<0.001), oral+pharyngeal phase (p=0.022), and oral + pharyngeal + esophageal phase (p=0.048) (Table 4).

  • Forth, the distribution of stroke locations is significantly different between PSCI patients with and without PSD. It is known that the prevalence of dysphagia is different depending on the stroke location (Dysphagia 2017 Dec;32(6):777-784). Therefore, Chi-square test should be conducted to examine whether there was a difference in the prevalence of dysphagia between patients with supratentorial stroke and those with infratentorial stroke.

Authors’ response: Thanks for your suggestion, the Chi-square test have been conducted to examine whether there was a difference in the prevalence of dysphagia between patients with supratentorial stroke and those with infratentorial stroke in Table 2. The results showed that there was a significant difference in the prevalence of dysphagia between patients with supratentorial stroke and infratentorial stroke.

Minor comments:

  • Professional English editing throughout the manuscript is required. In particular, lines 107-111 and lines 136-137 do not make sense.

Authors’ response: We thank the reviewer for the valuable suggestion, and the entire manuscript has been carefully checked by our group members and further edited by professional English editors.

  • No VFSS and PAS data were shown.

Authors’ response: Thanks for your suggestion, the VFSS and PAS data has been exhibited at Figure 2 and Table 3.

  • Ref 15 has a typo (fysphagia).

Authors’ response: Thanks for your suggestion, we have revised the manuscript.

Reviewer 2 Report

ABSTRACT

-       Good overall summary of the study. 

INTRODUCTION

-       Lines 37-38: Some concern arises over how cognitive impairment is described. The entity of PSCI is well presented and overall cognitive function well-defined, however, it should be made clear to the reader that “language” can be a part of cognitive impairment when in combination with other cognitive symptoms OR “language” can be a stand alone impairment post-stroke, and NOT considered a cognitive impairment. For example, in the case of a focal brain lesion in the dominant hemisphere, the patient may present with an Aphasia (language disorder). This is not classified as a cognitive impairment. 

-       In addition, the authors are cautioned to address the reality of the difficulties in testing for cognition in the presence of an Aphasia, or true language disorder.

-       Line 40: a better word than “improve” is “increase”

-       Line 42: Mention should be made that the incidence of PSD changes. During the first week post-stroke PDS is more prevalent; by the end of the first week the incidence of PSD declines. 

-       Line 47, page 2: central pattern generator, not center

MATERIALS AND METHODS

-       The inclusion criteria of 2 – 6 months seems a bit too broad. The authors should provide a rationale for this choice. 

-       Again, the manner in which cognitive impairment is determined is of concern. If the patient presents with a Fluent Aphasia, for example, he/she may not be able to complete the MoCA or the MMSE – and yet not be cognitively impaired. In Fluent Aphasia the patient’s language comprehension is impaired (can be auditory or visual or both).

-       Line 107, page 3: More information should be provided on VFSS: how recorded, frame rate, swallow materials/consistencies presented for evaluation.

-       PAS information: which swallow was assessed? Were all the swallows for one consistency averaged or was the more severe of the trials reported/analyzed?

-       Line 117, page 3: dysarthria is provided as an example of risk factors for analysis, however, neurologic disease is listed as an exclusion criteria item above. This should be clarified. 

RESULTS

-       All the subgroups appear unclear. It is unclear why now the term “executive function” is used for sub-dividing Group 1. Not clear on the difference between this group and the PSCI group. 

-       Also, why are “incomplete records” being mentioned? Were these not excluded from the study? 

-       Information reported in section 3.1 is very relevant. This makes a bit more sense. 

-       How the different phases of swallow were separated should be reported. For example, there is great debate about where the oral phase ends and the pharyngeal phase begins. Is premature spillage to the pyriform sinuses a dysfunction of the oral phase or of the pharyngeal phase, for example? 

-       Line 170, page 6: Unclear on the FOIS scores difference between admission and discharge. 

-       An important element is missing: a description of the actual biomechanical differences seen by sub-group in oral vs pharyngeal phase.

DISCUSSION

-       The discussion states that more oral stage dysphagia is seen in patients PSCI. Further data/support needed. 

GENERAL COMMENTS

-       This is a good study and required many hours of analysis and decision-making. It is an interesting area of study.

-       The study in its present form is not ready. Based on the comments above, edits should be made. 

-       It is this reviewer’s belief that with the proper edits, this will be a good addition to our current literature. 

Author Response

Reviewer 2:

ABSTRACT

Good overall summary of the study.

INTRODUCTION

  • Lines 37-38: Some concern arises over how cognitive impairment is described. The entity of PSCI is well presented and overall cognitive function well-defined, however, it should be made clear to the reader that “language” can be a part of cognitive impairment when in combination with other cognitive symptoms OR “language” can be a stand alone impairment post-stroke, and NOT considered a cognitive impairment. For example, in the case of a focal brain lesion in the dominant hemisphere, the patient may present with an Aphasia (language disorder). This is not classified as a cognitive impairment.

Authors’ response: Thanks for your suggestion. We have revised the introduction part (Page 3, Line 53-54): It is noteworthy that language impairment can be an isolated symptom after stroke, which is associated with damage of language centers.  

  • In addition, the authors are cautioned to address the reality of the difficulties in testing for cognition in the presence of an Aphasia, or true language disorder.

Authors’ response: I'm sorry we didn't explain that clearly. Patients with fluent aphasia (sensory aphasia) after stroke unable to finish the MMSE evaluation were excluded in the present study, while for patients with motor aphasia and cognitive impairment, the MMSE was used to evaluate the cognitive function and the score related to the language part was deducted. The present study is a retrospective cohort study based on the database, further study should choose the suitable assessment tools for a comprehensive assessment of PSCI patients with aphasia after stroke.

We have revised the introduction part of the manuscript as follows (Page 3, Line 50-52): Sometimes, language impairment can be an isolated symptom after stroke. The common assessment tools for PSCI include Mini Mental State Examination (MMSE), Montreal Cognitive Assessment (MoCA), and Cognitive Assessment for Stroke Patients (CASP). For patients with language impairment, the efficiency of CASP is higher than MMSE and MoCA.

For patients with aphasia after stroke, the efficiency of Mini–Mental State Examination (MMSE) was influenced according to the literature. The efficiency of the Assessment for Stroke Patients (CASP) scale was higher than the MMSE and Montreal Cognitive Assessment (MoCA).[1] Another study adopted MMSE to explore the difference between aphasia and dementia patients, the test results demonstrated that there were no significant differences between the two groups of participants' MMSE results.[2] Therefore, the efficiency of MMSE to aphasia remind uncertain.  

Reference:

  • Crivelli D, Spinosa C, Angelillo MT, Balconi M. The influence of language comprehension proficiency on assessment of global cognitive impairment following Acquired Brain Injury: A comparison between MMSE, MoCA and CASP batteries. Appl Neuropsychol Adult. 2021 Aug 21:1-6.
  • Myrberg K, Hydén LC, Samuelsson C. The mini-mental state examination (MMSE) from a language perspective: an analysis of test interaction. Clin Linguist Phon. 2020 Jul 2;34(7):652-670.

  • Line 40: a better word than “improve” is “increase”

Authors’ response: Thanks for your suggestion, we have revised the manuscript.

  • Line 42: Mention should be made that the incidence of PSD changes. During the first week post-stroke PDS is more prevalent; by the end of the first week the incidence of PSD declines.

Authors’ response: Thanks for your suggestion, the incidence rate of PSD varies from different reports. During the first week post-stroke PDS is more prevalent, and it’s incidence rate is 67%,[1] while in 2 weeks and 5 weeks is 45% and 37%.[2,3] We have revised the manuscript (Page 3, Line 58-60): Post-stroke dysphagia (PSD), another common complication after stroke, is closely associated with aspiration, pneumonia, malnutrition and dehydration, with a high estimated prevalence in the first week.

Reference:

[1] Hinds NP, Wiles CM. Assessment of swallowing and referral to speech and language therapists in acute stroke. QJM. 1998, 91:829-835.

[2 ]DePippo KL, Holas MA, Reding MJ. The burke dysphagia screening test: validation of its use in patients with stroke. Arch Phys Med Rehabil. 1994, 75:1284-1286.

[3] Gordon C, Hewer RL, Wade DT. Dysphagia in acute stroke. BMJ.1987, 295:411-414。

5.Line 47, page 2: central pattern generator, not center

Authors’ response: Thanks for your suggestion, we have revised the manuscript.

MATERIALS AND METHODS

  • The inclusion criteria of 2-6 months seems a bit too broad. The authors should provide a rationale for this choice.

Authors’ response: Thanks for your suggestion. The patients included in the present study were subacute phase stroke patients (less than 6 months after stroke onset).[1] The reasons we chose patients diagnosed with stroke between 2 weeks and 6 months are as follows: firstly, the choice was according to the reference;[2] secondly, the data was collated from the rehabilitation-specific disease database, and the patients admitted to the hospital are basically in the subacute phase of stroke.

  However, the inclusion criteria of 2 weeks to 6 months seems a bit too broad. The swallowing function of patients with PSD probably recovers spontaneously in the early stage of stroke, which may influence the results. Therefore, this is one of the limitations of the present study, and further study should reduce the time range of the onset of stroke to avoid bias.

Reference:

[1] Bernhardt, J., Hayward, K.S., Kwakkel, G., Ward, N.S., Wolf, S.L., Borschmann, K., Krakauer, J.W., Boyd, L.A., Carmichael, S.T., Corbett, D., Cramer, S.C., 2017. Agreed definitions and a shared vision for new standards in stroke recovery research: The Stroke Recovery and Rehabilitation Roundtable taskforce. Int. J. Stroke Off. J. Int. Stroke Soc. 12, 444-450.

[2] Bath PM, Lee HS, Everton LF. Swallowing therapy for dysphagia in acute and subacute stroke. Cochrane Database Syst Rev. 2018,10(10):CD000323.

  • Again, the manner in which cognitive impairment is determined is of concern. If the patient presents with a Fluent Aphasia, for example, he/she may not be able to complete the MoCA or the MMSE – and yet not be cognitively impaired. In Fluent Aphasia the patient’s language comprehension is impaired (can be auditory or visual or both).

Authors’ response: Thanks for your suggestion. Patients with fluent aphasia or sensory aphasia who can not be able to complete the MoCA or the MMSE were excluded from the present study. While patients with PSCI and motor aphasia were included, and MMSE was adopted to evaluate the cognitive function, the score related to the language part was deducted. The writing board was used to complete the evaluation process for patients with motor aphasia. The present study is a retrospective cohort study based on the database, the data is limited. Suitable assessment methods should be applied for patients with fluent aphasia and PSCI in further study to avoid bias. We have added these to the exclusion criteria in the revised manuscript (Page 4, Line 103-104): 4) patients with fluent aphasia and can not complete the MMSE or MoCA assessment.

3.Line 107, page 3: More information should be provided on VFSS: how recorded, frame rate, swallow materials/consistencies presented for evaluation.

Authors’ response: I'm sorry we didn't explain that clearly. We have revised the method part of the manuscript as follows (Page 5, Line 122-134): The VFSS examination was evaluated at admission by a speech-language therapist. During the VFSS, radiation exposure ranged up to the top of the nasal cavity, down to the C7 cervical spine, anterior and posterior to the lips and back of the neck. Patients received 3ml, 5ml, and 10ml of thickened and diluted barium liquid (contrast media: 60%w/v barium sulfate suspension), and the steps were performed according to the modified Logemann protocol. The video was analyzed by a well-trained physiotherapist. The various phases of swallowing and PAS score were recorded. The phases of swallowing including the oral phase (oral preparatory phase and oral propulsive phase), pharyngeal phase, and esophageal phase, and dysfunction can exist at any of the 4 phases of the physiologic swallow, and the definition of different phases were shown in Table 1 and Fig. 1. PAS was also scored by VFSS, and the highest score was recorded during the examination. PAS is an 8-point scale used to determine dysphagia severity. The higher the score, the more severe the penetration and aspiration symptoms.

4.PAS information: which swallow was assessed? Were all the swallows for one consistency averaged or was the more severe of the trials reported/analyzed?

Authors’ response: I'm sorry we didn't explain that clearly. We have revised the method part of the manuscript (Page 5, Line 132-134): PAS was also scored by VFSS, and the highest score was recorded during the examination. PAS is an 8-point scale used to determine dysphagia severity. The higher the score, the more severe the penetration and aspiration symptoms.

  • Line 117, page 3: dysarthria is provided as an example of risk factors for analysis, however, neurologic disease is listed as an exclusion criteria item above. This should be clarified.

Authors’ response: Thanks for your suggestion. Dysarthria is a speech disorder linked to difficulties controlling the muscles needed for speaking and caused by neurological impairment, such as cerebral palsy, stroke, traumatic brain injury, or neurological disease.[1] The dysarthria in the present study is caused by stroke instead of other neurologic diseases. Therefore, the neurologic disease is listed as an exclusion criterion in the present study.

Reference:

[1] Pennington L, Parker NK, Kelly H, Miller N. Speech therapy for children with dysarthria acquired before three years of age. Cochrane Database Syst Rev. 2016,7(7):CD006937.

RESULTS

  • All the subgroups appear unclear. It is unclear why now the term “executive function” is used for sub-dividing Group 1. Not clear on the difference between this group and the PSCI group.

Authors’ response: To avoid ambiguity and make the subgroup analysis clear, we have deleted the part of the subgroup analysis according to “executive function” and reanalyzed it according to the severity of PSCI, because the executive dysfunction is involved in PSCI.

  • Also, why are “incomplete records” being mentioned? Were these not excluded from the study?

Authors’ response: II'm sorry we didn't explain that clearly. We first analyze the incidence rate between PSCI and severe PSD based on the Pearson chi-square test. Then, the patients with incomplete records were excluded, a total of 331 patients were included in the final analysis to explore the relationship between PSCI and severe PSD.

We have revised the result part of the manuscript (Page 7, Line 166-168): Patients with incomplete medical records were excluded. Finally, a total of 331 were included for the analysis of risk factors, and 154 patients were included for further subgroup analysis based on the severity of PSCI.

3.Information reported in section 3.1 is very relevant. This makes a bit more sense.

Authors’ response: Thank you for your suggestion, we have revised the relevant content.

4.How the different phases of swallow were separated should be reported. For example, there is great debate about where the oral phase ends and the pharyngeal phase begins. Is premature spillage to the pyriform sinuses a dysfunction of the oral phase or of the pharyngeal phase, for example?

Authors’ response: I'm sorry we didn't explain that clearly. The oral phase included oral preparatory phase and oral propulsive phase. The oral preparatory phase refers to the food enters and processes in the oral cavity, and the oral propulsive phase refers to the food ball is formed after processing, and the food ball is transported to the pharyngeal. Therefore, oral phase dysphagia including excessive saliva and retention of food in the mouth, regurgitation into the nose, and difficulty in chewing food and leaking of food from the mouth.1 Furthermore, the premature spillage to the pyriform sinuses is a special dysfunction form of the oral phase, and caused by poor oral control and coordination.[2,3]

  The different phases of swallow were separated have been reported in the manuscript (Table1).

Table 1. The start and end point of different phases of swallowing.  

Phases of swallowing

Start point

End point

Oral preparatory phase

The food was processed in the oral cavity

Oral propulsive phase

The tongue voluntary presses the collected bolus/saliva against the palate

Bolus passing the ramus of the mandible

Pharyngeal phase

Bolus passing the ramus of the mandible

The upper esophageal sphincter relaxation

Esophageal phase

The upper esophageal sphincter relaxation

The bolus into the stomach

Reference:

[1] Milewska M, Grabarczyk K, DÄ…browska-Bender M, Jamróz B, Dziewulska D, Staniszewska A, Panczyk M, Szostak-WÄ™gierek D. The prevalence and types of oral- and pharyngeal-stage dysphagia in patients with demyelinating diseases based on subjective assessment by the study subjects. Mult Scler Relat Disord. 2020, 37:101484.

[2] Logemann J A . Dysphagia: Evaluation and Treatment[J]. Folia Phoniatr Logop, 1995.

[3] Dellavia C, Rosati R, Musto F, Pellegrini G, Begnoni G, Ferrario VF. Preliminary approach for the surface electromyographical evaluation of the oral phase of swallowing. J Oral Rehabil. 2018, 45(7):518-525.

  • Line 170, page 6: Unclear on the FOIS scores difference between admission and discharge.

Authors’ response: I'm sorry we didn't explain that clearly. The FOIS was adopted by the speech-language pathologist to evaluate the swallowing function not only at admission but at discharge to assess the curative effect. The FOIS change in the present study refers to the difference in FOIS between admission and discharge, which represent the curative effect of swallowing therapy during hospitalization.

  • An important element is missing: a description of the actual biomechanical differences seen by sub-group in oral vs pharyngeal phase.

Authors’ response: Thanks for your suggestion. Swallowing is a neuromuscular process that involves a complex sequence of sensorimotor events, which are executed to efficiently and safely transport food and liquid from the mouth to the stomach. Safe oropharyngeal swallowing involves the activation, modulation, and coordination of oral, pharyngeal, laryngeal, and esophageal structures and musculature. While the tongue and mandibular muscle play a key role in the regulation of swallowing function, especially in the oral preparatory phase and oral propulsive phase.[1] During the oral phase, masticatory muscles for jaw closing (masseter, temporalis and pterygoid muscles) and opening (submental muscles) are involved in order to stabilize the mandible during oral closure and elevate the tongue.[2] However, the executive dysfunction may interfere with the preparatory function of swallowing including food entering and processing in the oral cavity.[3,4] The results in the present study showed that severe PSCI patients manifest as oral phase dysfunction, while mild PSCI manifest as pharyngeal phase. This may cause biomechanical differences between the oral phase and pharyngeal phase. But the data was limited in the present study, further research is needed to explore the biomechanical differences between different swallowing phases.

Reference:

[1] Cuellar ME, Oommen E. Objective physiological measures of lingual and jaw function in healthy individuals and individuals with dysphagia due to neurodegenerative diseases. MethodsX. 2021, 8:101461.

[2] Dellavia C, Rosati R, Musto F, Pellegrini G, Begnoni G, Ferrario VF. Preliminary approach for the surface electromyographical evaluation of the oral phase of swallowing. J Oral Rehabil. 2018, 45(7):518-525.

[3] Malandraki GA, Sutton BP, Perlman A et al. Neural activation of swallowing and swallowing-related tasks in healthy young adults: An attempt to separate the components of deglutition. Hum Brain Mapp 2009,30:3209–3226.

[4] Reuter-Lorenz PA, Jonides J, Smith EE et al. Age differences in the frontal lateralization of verbal and spatial working memory revealed by PET. J Cogn Neurosci, 2000,12:174–187.

DISCUSSION

  • The discussion states that more oral stage dysphagia is seen in patients PSCI. Further data/support needed.

Authors’ response: The present study divided PSCI patients into mild, moderate, and severe PSCI subgroup, the results showed that severe PSCI patients manifest as oral phase dysfunction, while mild PSCI manifest as pharyngeal phase. The former studies believed that patients with executive dysfunction manifested as oral stage dysphagia,[1] which supported the validity of our original work.

The present study is a retrospective cohort study based on the database to investigate the relationship between post-stroke cognitive impairment (PSCI) and severe post-stroke dysphagia (PSD). It’s a preliminary result due to the limited data in the present study, and further high-quality study is needed to verify our results.

Reference:

[1] Yang EJ, Kim KW, Lim JY, et al. Relationship between dysphagia and mild cognitive impairment in a community-based elderly cohort: the korean longitudinal study on health and aging. Journal of the American Geriatrics Society, 2014, 62(1): 40-46.

Round 2

Reviewer 1 Report

The authors responded to all my comments, and the manuscript has improved significantly.

Reviewer 2 Report

Thank you for the fine revisions. All appear to be acceptable and improve the quality of the paper. Your work is appreciated.